# Heat Shock Protein 70 Is Involved in the Efficiency of Preconditioning with Cyclosporine A in Renal Ischemia Reperfusion Injury by Modulating Mitochondrial Functions

**DOI:** 10.3390/ijms24119541

**Published:** 2023-05-31

**Authors:** Maxime Schleef, Margaux Rozes, Bruno Pillot, Gabriel Bidaux, Fitsum Guebre-Egziabher, Laurent Juillard, Delphine Baetz, Sandrine Lemoine

**Affiliations:** 1CarMeN Laboratory, Inserm U1060, INRA U1397, Université Claude Bernard Lyon 1, 69500 Bron, France; maxime.schleef@chu-lyon.fr (M.S.); margaux.rozes@chu-lyon.fr (M.R.); bruno.pillot@univ-lyon1.fr (B.P.); gabriel.bidaux@univ-lyon1.fr (G.B.); fitsum.guebre-egziabher@chu-lyon.fr (F.G.-E.); laurent.juillard@chu-lyon.fr (L.J.); delphine.baetz@univ-lyon1.fr (D.B.); 2Hospices Civils de Lyon, Médecine Intensive Réanimation, Hôpital Edouard Herriot, 69003 Lyon, France; 3Hospices Civils de Lyon, Néphrologie-HTA-Dialyse, Hôpital Edouard Herriot, 69003 Lyon, France; 4Hospices Civils de Lyon, Explorations Fonctionnelles Rénales, Hôpital Edouard Herriot, 69003 Lyon, France

**Keywords:** renal ischemia reperfusion, cyclosporine A, preconditioning, Hsp70, mitochondria, mitochondrial permeability transition pore

## Abstract

Cyclosporine A (CsA) preconditioning is known to target mitochondrial permeability transition pore and protect renal function after ischemia reperfusion (IR). The upregulation of heat-shock protein 70 (Hsp70) expression after CsA injection is thought to be associated with renal protection. The aim of this study was to test the effect of Hsp70 expression on kidney and mitochondria functions after IR. Mice underwent a right unilateral nephrectomy and 30 min of left renal artery clamping, performed after CsA injection and/or administration of the Hsp70 inhibitor. Histological score, plasma creatinine, mitochondrial calcium retention capacity, and oxidative phosphorylation were assessed after 24 h of reperfusion. In parallel, we used a model of hypoxia reoxygenation on HK2 cells to modulate Hsp70 expression using an SiRNA or a plasmid. We assessed cell death after 18 h of hypoxia and 4 h of reoxygenation. CsA significantly improved renal function, histological score, and mitochondrial functions compared to the ischemic group but the inhibition of Hsp70 repealed the protection afforded by CsA injection. In vitro, Hsp70 inhibition by SiRNA increased cell death. Conversely, Hsp70 overexpression protected cells from the hypoxic condition, as well as the CsA injection. We did not find a synergic association between Hsp70 expression and CsA use. We demonstrated Hsp70 could modulate mitochondrial functions to protect kidneys from IR. This pathway may be targeted by drugs to provide new therapeutics to improve renal function after IR.

## 1. Introduction

Mitochondria play a major role in the pathophysiology of ischemia reperfusion (IR). It is accepted now that the opening of a nonselective channel, called mitochondrial permeability transition pore (MPTP) after IR leads to cell death. Its opening at the time of reperfusion implies a matrix swelling, membrane potential collapse, uncoupled oxidative phosphorylation, ATP consumption, outer membrane rupture, cytochrome C release, and cell death [1]. Various strategies targeting MPTP during IR, called conditioning, have been proposed to inhibit its opening and protect cells from IR injury.

Among these strategies, cyclosporine A (CsA) has been previously tested in several organs submitted to IR. CsA binds a mitochondrial matrix soluble protein called cyclophilin D (Cyp-D) and delays the MPTP opening, protecting organs from cell death following IR. Although repeated and prolonged CsA intake leads to nephrotoxicity, previous experimental data obtained in our laboratory suggested a protective role of a single-dose CsA injection just before renal IR. We highlighted that the dose and timing of CsA preconditioning were also crucial to ensure an effective treatment [2].

The Hsp70 family encompasses a diverse group of so-called heat-shock proteins, with distinct cellular localizations, including the cytosol, endoplasmic reticulum, and mitochondria, and with distinct functions, including the folding, stabilization, and transport of other proteins. These highly conserved chaperone proteins play a crucial role in maintaining cellular homeostasis, particularly under conditions of environmental stress [3]. Among the different Hsp70 proteins, there are four major groups of proteins with a 70 kDa molecular weight. Two of them are the stress-inducible Hsp70-like HSPA1A and HSPA1B commonly named Hsp70, and the endoplasmic reticulum (ER)-localized Grp78 (HSPA5), and some are constitutively expressed as Hsc70 (HSPA8) and mortalin/GRP75 (HSPA9) in the mitochondria [4]. A recent focus has also emerged on the dual role of extracellular Hsp70, which rather acts as a deleterious proinflammatory mediator when released from injured cells [5]. In the present study, we chose to investigate the role of cytoplasmic inducible Hsp70 as its role in ischemia was already described [6].

Inducible Hsp70 is notably known to be involved in cell survival [3,7] and to confer cytoprotection by inhibiting cell death in stress conditions [8]. The role of mitochondrial Hsp70 as a key factor in the formation of mitochondrial protein machinery has also recently been highlighted [9]. It has been previously shown that the overexpression of Hsp70 protects renal cells from oxidative stress, notably through an action on mitochondria [10,11,12]. Hsp70 overexpression by glutamine significantly reduced myoglobinuric acute kidney injury by limiting c-junNH2 amino-terminal kinases (JNK) activation and apoptosis [13]. Moreover, an induction of Hsp70 expression by geranylgeranylacetone significantly blunted renal ischemic tubular injury in mice [14]. In parallel, older studies reported an upregulation of Hsp70 expression after CsA injection that was associated with a nephroprotection [15]. Moreover, the addition of an Hsp70 inhibitor, quercetin [16], reverted CsA’s protective effects on renal function [17].

Therefore, we hypothesized that a CsA preconditioning effect on renal preservation could imply an overexpression of inducible Hsp70 in the MPTP modulation pathway. Thus, we aimed to test the effect of the modulation of Hsp70 expression on kidney function and on MPTP opening, in the setting of renal IR in vivo and in vitro.

## 2. Results

### 2.1. CsA Regulates Hsp70 Level In Vitro and In Vivo

In vivo, CsA was associated with a significant upregulation of Hsp70 expression in the kidney 1 hour after injection (i.e., at 20 min of reperfusion) when applied before renal ischemia at a dose of 10 mg/kg (ratio 1.48 [1.29–1.84]) compared to the ischemic group (ratio 0.95 [0.9–1.17]; *p* = 0.02) in mice kidney tissue (Figure 1A). However, after 24 h of reperfusion, no significant difference persisted (Figure 1B). In vitro, we showed that 0.5 µM of CsA applied at the beginning of oxygen–glucose deprivation (OGD) significantly increased the relative Hsp70 cellular expression (*p* = 0.017) when compared to the hypoxic condition after 18 h of OGD and 4 h of reoxygenation in HK2 cells (Figure 1C).

### 2.2. Hsp70 Inhibition by Quercetin Decreased Renal Protective Effect of CsA In Vivo and In Vitro

In vivo, after 24 h of reperfusion, serum creatinine significantly increased in the ischemic group (1.6 [1.46–2.12] mg/dL) compared to the sham group (0.14 [0.14–0.14] mg/dL; *p* = 0.02). The increase in creatinine was significantly blunted in the precond-CsA 10 mg/kg group (0.88 [0.87–0.93] mg/dL) compared to the ischemic group (*p* = 0.002).

In order to test the protective effect of the Hsp70 increase by CsA on renal function, we used an inhibitor of Hsp70, quercetin, 2 h before ischemia at the dose of 100 mg/kg. Quercetin inhibited CsA’s protective effect on serum creatinine (1.4 [1.37–1.75] mg/dL) when compared to CsA preconditioning alone (*p* = 0.004). The use of quercetin alone in ischemic mice (1.49 [1.43–1.51] mg/dL) did not affect plasma creatinine level when compared to the ischemic group (*p* = 0.25; Figure 2A). Moreover, serum creatinine level did not differ between sham mice and sham mice exposed to quercetin.

The histological score of acute tubular necrosis (i.e., renal IR injury) was also significantly improved in the precond-CsA 10 mg/kg group (score 1.3 [1.2–2.2]) when compared to the ischemic group (score 2.6 [2.5–3.5]; *p* = 0.002). In the presence of quercetin, CsA was no longer able to preserve the histological score (precond-CsA 0.5 µM + Q; score 3.0 [3.0–3.5]; *p* = 0.002) (Figure 2B).

We performed similar experiments in vitro on HK2 cells. CsA preconditioning significantly decreased cell death (−18%) after 18 h of OGD and 4 h of reoxygenation compared to the hypoxic condition (*p* = 0.008). Quercetin attenuated the protective effect of CsA on cell death versus precond-CsA 0.5 µM (*p* = 0.05). Quercetin alone did not affect cell death compared to hypoxic condition (*p* = 0.11) (Figure 2C). We also found, as historically described [16], that quercetin seemed to inhibit Hsp70 by reducing its protein expression in HK2 cells undergoing OGD and reoxygenation, as well as its overexpression after CsA preconditioning (Figure 3).

In order to confirm the role of Hsp70 in the protection afforded by CsA, we modulated the expression of inducible Hsp70 using an SiRNA against Hsp70, or an Hsp70-plasmid. We first validated that adding SiRNA against HSP70 on HK2 cells effectively decreased Hsp70 protein expression compared to a scramble SiRNA, and that adding an Hsp70 plasmid tended to increase Hsp70 protein expression compared to an empty plasmid (Appendix A).

The presence of a scramble SiRNA did not significantly modified the effect of 0.5 µM CsA on cell death. We still observed a significant decrease of cell death after CsA preconditioning (HR SiRNA CTL vs. HR SiRNA CTL + CsA; *p* < 0.0001). As expected, with a genetic repression of inducible Hsp70 expression using SiRNA directed against HSP70, CsA no longer protected against cell death (HR SiRNA HSP70 + CsA vs. HR SiRNA CTL; *p* = 0.14), which was significantly higher compared to CsA preconditioning alone (HR SiRNA HSP70 + CsA vs. HR SiRNA CTL + CsA; *p* = 0.001). Hsp70’s blunted expression alone did not impact cell death after H/R compared to the SiRNA CTL condition (Figure 4A).

The induction of a plasmid-mediated overexpression of Hsp70 provided a protection against HR-induced cell death, when compared to a control empty plasmid (*p* = 0.017) as CsA alone. The adjunction of CsA did not show any additive protection when compared to plasmid–Hsp70 overexpression only (Figure 4B).

### 2.3. Effect of Pharmacological Hsp70 Inhibition after IR on Kidney Mitochondrial Functions

We assessed CRC and mitochondrial respiration (oxidative phosphorylation) in vivo to test the effect of the modulation of Hsp70 expression by CsA preconditioning on the protection of mitochondrial functions.

The amount of Ca^2+^ required to trigger a massive Ca^2+^ release by mitochondria into the cytosol (i.e., CRC) tended to be reduced in the ischemic group after 24 h of reperfusion, when compared to the sham group (110 [40–160] and 360 [160–440] nmol Ca^2+^/mg proteins, respectively; *p* = 0.06). CRC significantly increased after preconditioning with CsA (240 [220–300] nmol Ca^2+^/mg proteins) when compared to the ischemic group (*p* = 0.002). When quercetin was added to CsA preconditioning, CsA no longer had any protective effect on CRC (80 [40–80] nmol Ca^2+^/mg proteins; *p* = 0.008; Figure 5A).

After 24 h of reperfusion, there was a tendency towards an increased maximal mitochondrial consumption of oxygen with ADP (i.e., state 3) in the precond-CsA 10 mg/kg group (50.5 [39.3–72.3] nmol O_2_/min/mg proteins), as compared to the ischemic group (29 [22–52.8] nmol O_2_/min/mg proteins; *p* = 0.1). More importantly, the use of quercetin with CsA significantly decreased state 3 mitochondrial respiration (16 [13.9–22.2] nmol O_2_/min/mg proteins) after 24 h of reperfusion when compared to the use of CsA alone (*p* = 0.004; Figure 5B). In parallel, oxygen consumption after completion of ADP phosphorylation (state 4) decreased when quercetin was added to CsA preconditioning (4.9 [4.3–7.4] nmol O_2_/min/mg proteins) as compared to CsA preconditioning alone (16.5 [13.2–19] nmol O_2_/min/mg proteins; *p* = 0.004; Figure 5C).

## 3. Discussion

In the present study, we showed that the nephroprotection by CsA preconditioning was partly mediated by Hsp70 overexpression and through MPTP opening modulation.

We confirmed that CsA could induce Hsp70 overexpression early after injection, since we were able to show a significant increase of its expression in vivo as soon as 1 h after CsA injection (which corresponded to the timing of 20 min of reperfusion in our model). Moreover, this effect was the same in HK2 cells pretreated with CsA. It has already been demonstrated that pretreating cells with low dose of CsA (<1 µM) exerts a preconditioning effect by generating a small number of ROS, leading to Hsp70 mRNA expression and fast Hsp70 protein expression [18]. Moreover, other authors showed that Hsp72 was detected in apical proximal tubules of the kidney after IR, where morphological alterations were the most important 15 min after reperfusion [19]. A fast mRNA and protein induction could account for the nephroprotective effect we reported herein, since hypoxia began once Hsp70 had already been overexpressed. However, there was no more difference of Hsp70 expression after 24 h of reperfusion. This can be explained by the fact that hypoxia per se leads to Hsp70 expression, which was delayed compared to the earlier CsA preconditioning.

We showed that Hsp70 overexpression was associated with a better cell viability and renal function following IR or OGD. The importance of Hsp70 in renal protection has already been supported by the increased death rate in Hsp70 knocked-out mice exposed to IR [14]. Yang et al. also showed that the administration of CsA before ischemia increased Hsp70 expression with renal function and histology improvements [15], and pharmacological preconditioning with geranylgeranylacetone (an Hsp70 inducer) has been shown to improve renal function after IR [20]. Although MPTP opening is thought to be regulated by CsA through its binding to CypD, our results support the hypothesis that CsA could also modulate MPTP via another Hsp70-dependent signaling pathway as previously suggested [17]. In the present study, we reported that in vivo pharmacological Hsp70 inhibition abrogated CsA protection from renal IR injury. Indeed, we used quercetin to inhibit Hsp70 in in vivo experiments. Quercetin interacts with heat-shock factors and inhibits the transcriptional activation of heat-shock proteins’ genes [21], notably decreasing Hsp70 protein expression [16]. Quercetin is, however, not a specific Hsp70 inhibitor and is also known for its protective antioxidant, vasodilator, and anti-inflammatory properties. Nevertheless, in our experiments, quercetin was associated with a significant decrease of Hsp70 expression and led to the loss of the protective effects of CsA as expected. Moreover, using quercetin alone did not show any beneficial effects even on cellular model, which let us to speculate that its other protective properties were not involved. Therefore, in order to confirm the effect of inducible Hsp70, we chose to use an SiRNA against Hsp70 (HSPA1A and HSPA1B). We confirmed in vitro that the SiRNA against Hsp70 also abrogated the renal protection afforded by CsA, as quercetin did. Accordingly, we showed that Hsp70 overexpression alone protected renal cells from HR-induced cell death, as did CsA preconditioning. However, we did not find any additive synergic response of Hsp70 overexpression (by adding an Hsp70 plasmid) combined to CsA preconditioning, leading us to think that the protection conferred by CsA in our conditions should mainly involve this suggested Hsp70-dependent pathway.

We also studied herein the role of Hsp70 expression modulation on mitochondrial function. Hsp70 inhibition with quercetin before CsA injection worsened mitochondria CRC after 24 h of reperfusion, suggesting that MPTP opening was sensitive to Hsp70 expression. We showed an improvement of mitochondrial respiration (oxidative phosphorylation) after CsA injection, with a significant increase of maximal oxygen consumption with ADP (i.e., state 3) compared to the ischemic group. Accordingly, we were able to show that the inhibition of Hsp70 before CsA injection led to a drop of mitochondrial respiration, i.e., a loss of its protective effect. These results underscore a probable relationship between Hsp70 and MPTP. This is in line with previous results showing that the overexpression of Hsp70 improved the recovery of state 3 mitochondrial respiration and ATP content in renal proximal tubular cells [10]. Other previous experimental data indicated that the induction of Hsp70 was associated with a decrease of ROS production, the preservation of state 3 mitochondrial respiration, and the preservation of mitochondrial membrane potential in astrocytes [22]. Moreover, the overexpression of Hsp70 in transgenic mice was shown to protect the heart from IR injury via a mitochondrial protection [23].

The main limit of the present study is that the complete signaling pathway between Hsp70 and MPTP remains unclear. The protective mechanism of Hsp70 in renal IR injury remains poorly understood. It has been suggested that this protection could be mediated by the correction of protein conformation, cytoskeletal stabilization, ant-inflammatory effects, antiapoptotic properties, and the stimulation of regulatory T cells [3,7,24,25]. Herein, we examined the potential link between CsA preconditioning, Hsp70 expression and mitochondria/MPTP. However, the exact mechanisms of how Hsp70 can modulate MPTP opening remains to be determined. One can speculate that it may involve GSK-3β: Hsp70 has been shown to facilitate the inhibition of GSK-3β activity, known to regulate MPTP opening during IR [14]. Unfortunately, we were not able to measure GSK-3β expression or activity in vivo in the kidney or in vitro in HK2 cells herein. The role of oxidative stress in this setting could also be of interest, although we were limited by the amount of blood, kidney, and the number of mitochondria samples (owing to the “Reduction” ethics principle of minimizing the number of animals used) to assess oxidative stress markers in vivo. Of note, this limitation also prevented us from measuring other renal function biomarkers than creatinine; nevertheless, we assessed renal injuries most precisely with a score of renal histological lesions. Eventually, we focused in the present study on intracellular cytoplasmic Hsp70 expression. Extracellular Hsp70, whether secreted or released by injured cells, has been described as a damage-associated molecular pattern, associated with worse outcomes after cardiac IR [26], and its precise role in renal IR, mitochondrial dysfunction, and the nephroprotection conferred by CsA remains yet to be elucidated, as we could not study it herein.

In conclusion, we demonstrated that in the setting of renal IR, Hsp70 could modulate mitochondrial functions (oxidative phosphorylation and MPTP opening) to protect the kidney from IR injuries. This pathway may be targeted by drugs to provide new therapeutics to improve renal function after IR.

## 4. Materials and Methods

### 4.1. Animals

All experiments were performed in accordance with the principles of laboratory animal care and French law and were approved by our local ethics committee (Claude Bernard Lyon 1 University, number BH2012-81).

Eight- to ten-week-old male C57BL6 mice (Charles River, Écully, France) were used for the study. Anesthesia was performed by an intraperitoneal injection of xylazine (5 mg/kg, Rompun^®^, Bayer, Puteaux, France), ketamine (100 mg/kg, Imalgene^®^ 1000, Acyon, Melan, France), and bupremorphine (0.075 mg/kg, Vetergesic^®^, Sogeval, Laval, France). Core temperature was maintained at 37 °C using a homeothermic pallet unit controlled by a rectal thermometer. A 22-gauge catheter was used in order to ventilate the animals (Minivent, Harvard Apparatus, March, Germany) at a frequency of 140 per minute with a tidal volume set at 0.15 mL. Before the start of the left kidney ischemia, we performed a laparotomy and right nephrectomy. Then, by clamping the left renal vascular pedicle with a microvascular clamp (Roboz Surgical Instruments, Washington, DC, USA), a 30 min period of ischemia was conducted by selective clamping. The reperfusion was checked by a direct visualization of the recoloration of the ischemic kidney. Sham animals did not undergo right nephrectomy nor ischemia.

Five groups were created: sham group; ischemic group; preconditioning with 10 mg/kg of CsA (precond-CsA 10 mg/kg) [2]; ischemic + quercetin (ischemic + Q); preconditioning with 10 mg/kg of CsA + quercetin (precond-CsA 10 mg/kg + Q).

There were two reperfusion durations: 20 min and 24 h (Appendix A). Preconditioning consisted in the intravenous injection of CsA at 10 mg/kg, 10 min before ischemia. A reperfusion of 20 min corresponded to a timing of 1 h passed since the CsA injection (i.e., 10 min before injection, plus 30 min of ischemia, plus 20 min of reperfusion = 60 min) In the precond-CsA 10 mg/kg + Q and the ischemic + Q groups, 100 mg/kg of quercetin were administrated intraperitoneally 2 h before ischemia. Immediately after the surgical procedure, the mice were rehydrated with a subcutaneous injection of 1 mL of 0.9% saline. At the end of the follow-up, a retro-orbital puncture was performed under deep anesthesia to withdraw 250 µL of blood. The left kidney was rapidly removed and divided into three samples: one for isolation of fresh mitochondria, one for histology and one for immunoblot analysis. Then, mice were euthanized by cervical dislocation.

### 4.2. Renal Parameters

Twenty-four hours after reperfusion, acute renal failure quantification was performed by measuring the serum creatinine concentration. Renal tubular injury was scored with a semiquantitative scoring system ranking from 0 to 4 by a blinded pathologist who examined 10 fields or more (×200 magnification) of four-micrometer sections of kidney tissue stained with periodic acid–Schiff, after it had been fixed with formalin, as previously described [2,27,28].

### 4.3. Mitochondria Functions

#### 4.3.1. Preparation of Fresh Isolated Mitochondria

The excised kidney was immediately immersed in a cold isolation buffer (0.25 M sucrose, 2 mM EGTA, and 10 mM Tris, pH 7.4). The tissue was cut into small pieces and mixed with a Potter Elvehjem in 10 µL of isolation buffer/mg tissue. We centrifugated the homogenate at 600× *g* for 7 min and then the supernatant at 6400× *g* for 10 min. The pellet containing the mitochondria was resuspended and homogenized in the isolation buffer without EGTA. A Bradford assay was used to determine protein concentration.

#### 4.3.2. Calcium Retention Capacity

The calcium retention capacity (CRC) is the quantity of Ca^2+^ required to trigger a massive Ca^2+^ release by mitochondria (MPTP opening) into the cytosol, as described by Ichas et al. [29]. This amount of calcium required to open the MPTP was measured as previously described [2,27,28], expressed as nmol Ca^2+^/mg mitochondrial proteins, and it assessed the functional resistance of mitochondria to calcic overload stress stimuli.

#### 4.3.3. Oxidative Phosphorylation

Mitochondrial respiratory function was determined as previously described [2,27,28]. Oxygen consumption was expressed in nmol of oxygen/min/mg of protein. The state 3 represents the maximal oxygen consumption rate in the presence of ADP and all substrates; the state 4 represents the oxygen consumption rate without ADP.

### 4.4. Cell Culture

We used HK2 cells, an immortalized proximal tubule epithelial cell line from adult human kidney (ATCC^®^ CRL2190™). HK2 cells were cultured in DMEM-F12 (Dulbecco’s modified Eagle Medium) treated with fetal bovine serum (FBS 10%) and antibiotics (penicillin and streptomycin 1%) at 37 °C and 5% of CO_2_.

### 4.5. Overexpression and Silencing of Hsp70

To induce the overexpression of Hsp70, HK2 cells were transfected with a plasmid containing human *hspA1A/70* gene or an empty plasmid (Addgene, Watertown, NY, USA). pcDNA5/FRT/TO GFP HSPA1A was a gift from Harm Kampinga (Addgene plasmid #19483; http://n2t.net/addgene:19483 accessed on 27 April 2023; RRID:Addgene_19483) [30]. To silence Hsp70 expression, HK2 cells were transfected with a mixture of siRNA against mRNA of human *HSPA1A* and human *HSPA1B*, or a negative control siRNA (Sigma-Aldrich, Saint-Louis, MO, USA). Transfections were all performed with the help of DharmaFECT Duo (Horizon, Cambridge, UK) according to the manufacturer’s instructions.

### 4.6. Hypoxia Reoxygenation (HR): “Oxygen Glucose Deprivation”

Forty-height hours after plating, cells were submitted to oxygen and glucose deprivation (OGD) at 37 °C in a hypoxic incubator where the oxygen level was controlled at 0.5% in a Tyrode solution devoid of nutrient (130 mM NaCl, 5 mM KCl, 10 mM Hepes, 1 mM MgCl_2_, 1.8 mM CaCl_2_) after two washes of culture plates with the Tyrode solution. The optimal time of hypoxia and reoxygenation to obtain an adequate rate of lethal reperfusion injury was defined by time-lapse experiments. Cell reoxygenation was performed by adding a complete medium (DMEM-F12 + FBS 10% and antibiotics) onto the cells and placing the plate in a regular normoxic incubator.

### 4.7. Cell Death Assay

After 18 h of OGD and 4 h of reoxygenation, cells were detached from the plate using Accutase (PAA laboratories, Linz, Austria). Cell death was quantified by flow cytometry (fluorescence-activated cell sorting, FACS) with propidium iodide (PI) (1μg/mL; Invitrogen™, Waltham, MA, USA). Triplicates were made for each condition. In each condition, a total of 2 000 events were counted by the flow cytometer.

The effect of CsA on cell viability was evaluated with an infusion of 0.5 µM of CsA for one hour before hypoxia (precond-CsA 0.5 µM). The effect of the Hsp70 inhibition on cell viability was studied by using quercetin. Fifty micromolar of quercetin (Q) was infused for one hour before hypoxia (Appendix A).

Four conditions were defined after preliminary experiments on HK2 cells: hypoxic condition: cells submitted to hypoxia and reoxygenation; precond-CsA 0.5 µM condition: hypoxic cells with an infusion of 0.5 µM of CsA; hypoxic + quercetin condition (hypoxic + Q); precond-CsA 0.5 µM + quercetin condition (precond-CsA 0.5 µM + Q).

### 4.8. Immunoblot Analysis

Twenty-five to one hundred micrograms of denatured protein samples was separated by SDS-PAGE and transferred to polyvinylidene difluoride membranes. Blots were blocked with 5% of nonfat milk or BSA in Tris-buffered saline (TBS–tween 0.05%). Membranes were incubated with primary antibody against Hsp70 (monoclonal anti-Hsp70; BD transduction laboratories, Lexington, KY, USA) or GAPDH (rabbit monoclonal antibody; Santa Cruz, Dallas, TX, USA) overnight at 4 °C. Then, membranes were washed 3 times in TBS–tween 0.05% and incubated with horseradish peroxidase coupled to the appropriate secondary antibody solution for one hour at room temperature. The protein of interest was detected by chemiluminescence by using Amersham™ ECL™ Prime Western Blotting Detection Reagents (General Electric Healthcare Life Sciences^®^, Chicago, IL, USA). Chemiluminescence detection was done with ChemiDoc™ XRS + system from Bio-Rad™ (Hercules, CA, USA). Data were analyzed with computer software Image Lab 4.0 from Bio-Rad Laboratories©. The relative expression levels of Hsp70 were presented as ratios to the levels of GAPDH expression.

### 4.9. Statistical Analysis

Statistical analyses were performed with the Prism software package (version 6; GraphPad, Boston, MA, USA). Data are expressed as median with interquartile (IQR). Wilcoxon (matched-pairs data) and Mann–Whitney tests were performed for the comparisons between the two groups. Friedman (matched-pairs data) and Kruskal–Wallis tests were used for the comparisons between more than two groups. Differences with *p*-values less than 0.05 were considered statistically significant.

## 5. Conclusions

We demonstrated that in the setting of renal IR, Hsp70 could modulate mitochondrial functions (oxidative phosphorylation and MPTP opening) to protect the kidney from IR injuries. This pathway may be targeted by drugs to provide new therapeutics to improve renal function after IR.

## Figures and Tables

**Figure 1 ijms-24-09541-f001:**
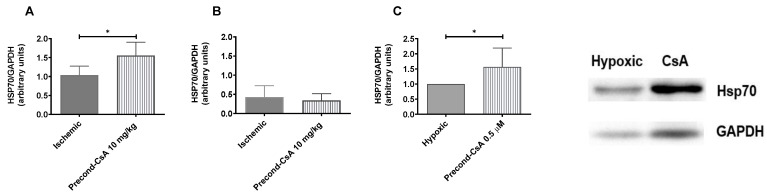
Effect of CsA preconditioning on Hsp70 expression: Western blot analysis. (**A**). Relative Hsp70 expression in mice kidneys increased in the precond-CsA 10 mg/kg group when compared to the ischemic group 20 min after reperfusion (*p* = 0.02). Ischemic, *n* = 6; precond-CsA 10 mg/kg, *n* = 6. (**B**). No difference between Hsp70 expression in mice kidney 24 h after reperfusion, *p* = 0.9. (**C**). Relative Hsp70 cellular expression of HK2 cells increased with CsA preconditioning when compared to oxygen–glucose deprivation (hypoxic) condition (*p* = 0.017). Hypoxic, *n* = 4 cell preparations; precond-CsA 0.5 µM, *n* = 3 cell preparations. Results are shown as medians with interquartiles. * *p* < 0.05, Mann–Whitney test.

**Figure 2 ijms-24-09541-f002:**
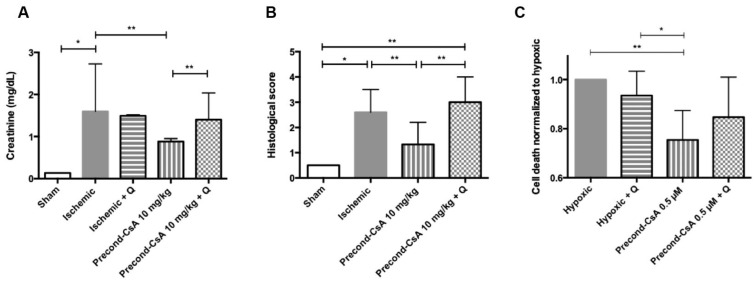
Effect of CsA preconditioning and quercetin on mice kidney function and histology. (**A**). Serum creatinine level. Sham, *n* = 3; ischemic, *n* = 7; ischemic + Q, *n* = 4; precond-CsA 10 mg/kg, *n* = 6; precond-CsA 10 mg/kg + Q, *n* = 5. (**B**). Histological score of acute tubular necrosis (ranging from 0 to 4). Sham, *n* = 3; ischemic, *n* = 6; precond-CsA 10 mg/kg, *n* = 6; precond-CsA 10 mg/kg + Q, *n* = 5. (**C**). Effect of CsA preconditioning and quercetin on HK2 cell death measured by propidium iodide staining, after 18 h of oxygen–glucose deprivation and 4 h of reoxygenation without any pretreatment (hypoxic, *n* = 4 cell preparations), after a 1 h infusion of 0.5 µM of CsA (precond-CsA 0.5 µM, *n* = 3 cell preparations), after a 1 h infusion of 50 µM of quercetin (hypoxic + Q, *n* = 2 cell preparations), or after a 1 h infusion of both CsA and quercetin (precond-CsA 0.5 µM + Q, *n* = 2 preparations). Triplicate samples were prepared for each condition and a total of 2000 events were acquired for each cell preparation (*n* = 8). Results are shown as medians with interquartiles. * *p* < 0.05, ** *p* < 0.01, Mann–Whitney test.

**Figure 3 ijms-24-09541-f003:**
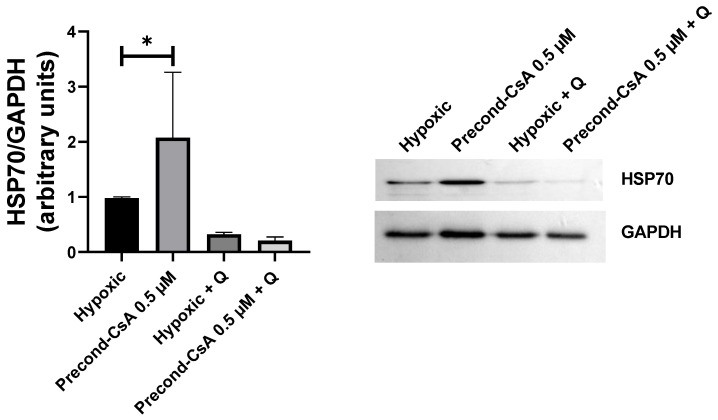
Effect of quercetin on Hsp70 protein expression in HK2 cells undergoing oxygen–glucose deprivation: Western blot analysis. HK2 cells were submitted to 18 h of oxygen–glucose deprivation and 4 h of reoxygenation without any pretreatment (hypoxic, *n* = 4 cell preparations), after a 1 h infusion of 0.5 µM of CsA (precond-CsA 0.5 µM, *n* = 3 cell preparations), after a 1 h infusion of 50 µM of quercetin (hypoxic + Q, *n* = 2 cell preparations), or after a 1 h infusion of both CsA and quercetin (precond-CsA 0.5 µM + Q, *n* = 2 preparations). Results are shown as medians with interquartiles. * *p* < 0.05, Mann–Whitney test.

**Figure 4 ijms-24-09541-f004:**
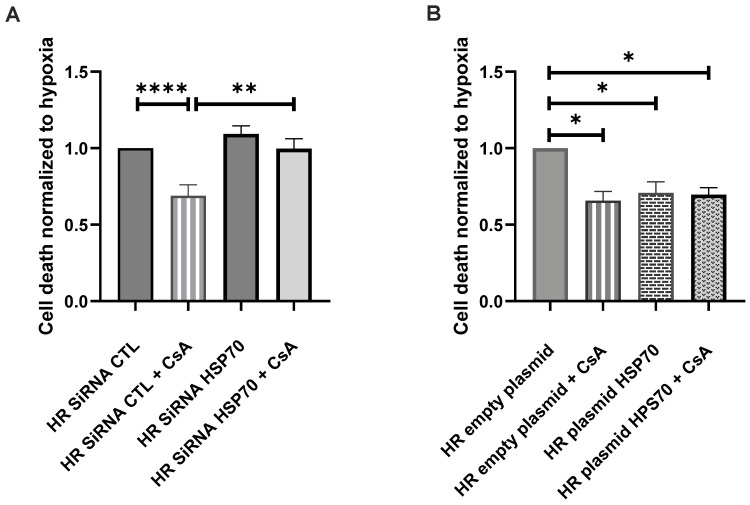
Effect of SiRNA Hsp70 and Hsp70 plasmid. (**A**). Hsp70 inhibition with an SiRNA. (**B**). Hsp70 overexpression with a plasmid. Results are shown as medians with interquartiles of *n* = 20 experiments for SiRNA Hsp70, and *n* = 7 experiments for Hsp70 plasmid. * *p* < 0.05, ** *p* < 0.01, **** *p* < 0.0001, Mann–Whitney test.

**Figure 5 ijms-24-09541-f005:**
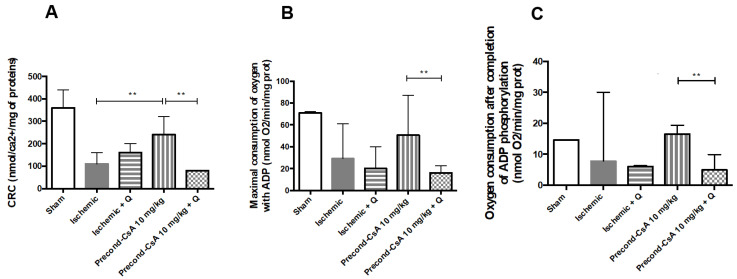
Effect of CsA preconditioning and quercetin on kidney mitochondrial functions. (**A**). Calcium retention capacity. Sham, *n* = 3; ischemic, *n* = 6; ischemic + Q, *n* = 3; precond-CsA 10 mg/kg, *n* = 5; precond-CsA 10 mg/kg + Q, *n* = 5. (**B**). State 3 of mitochondrial respiration (oxidative phosphorylation, OX-PHOS). Sham, *n* = 3; ischemic, *n* = 6; ischemic + Q, *n* = 3; precond-CsA 10 mg/kg, *n* = 6; precond-CsA 10 mg/kg + Q, *n* = 5. (**C**). State 4 of mitochondrial respiration (oxidative phosphorylation, OX-PHOS). Sham, *n* = 3; ischemic, *n* = 6; ischemic + Q, *n* = 3; precond-CsA 10 mg/kg, *n* = 6; precond-CsA 10 mg/kg + Q, *n* = 5. ** *p* < 0.01, Mann–Whitney test.

## Data Availability

The data presented in this study are available on request from the corresponding authors.

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
