# Peer review of "Heat Shock Protein 70 Is Involved in the Efficiency of Preconditioning with Cyclosporine A in Renal Ischemia Reperfusion Injury by Modulating Mitochondrial Functions"

_ijms, 2023, doi:10.3390/ijms24119541_

Round 1

Reviewer 1 Report

The selected topic is interesting. In below, I listed some comments that I think can help authors to improve their manuscript:

Minor comments

1.   The introduction need to be more clear and I think a lot of sentences need to remove to the discussion part.

2.   The kidney function markers like BUN and also NGAL need to be represented in the results

3.   Oxidative stress markers  need to be measured.

4.   Conclusion need to more obvious.

5.   The plagiarism is 31% ..The Manuscript need more edit.

Reviewer 2 Report

The manuscript presents the results on the role of stress-inducible cytoplasmic HSP70 in renal ischemia reperfusion injury. These are interesting results, however, there are several major issues:

1. The authors should provide a short paragraph in the Introduction on HSP70 family

2. The authors have not measured the role of extracellular HSP70 ,eventhough extracellular HSPs are currently of interest in the field. The information on extracellular HSPs was also not provided in the Introduction

3. Quercetin is not a selective HSP70 inhibitor, it also does not inhibit HSP72 homolog, why authors used it? Authors should provide the rationale on the use of Quercetin instead of selective HSP70 inhibitors used by other researchers 

4. Since authors showed that it affects mitochondrial dysfunction, why authors didn't explore the role of mitochondrial HSP70 ( mortalin)? 

Round 2

Reviewer 2 Report

The authors have addressed the comments